# Evidence from a long-term experiment that collective risks change social norms and promote cooperation

Aron Szekely [1,2,3,10]✉, Francesca Lipari [4,10], Alberto Antonioni [4], Mario Paolucci [1,5], Angel Sánchez [4,6,7,8], Luca Tummolini [1,2] & Giulia Andrighetto[1,2,9]

Social norms can help solve pressing societal challenges, from mitigating climate change to reducing the spread of infectious diseases. Despite their relevance, how norms shape cooperation among strangers remains insufficiently understood. Influential theories also suggest that the level of threat faced by different societies plays a key role in the strength of the norms that cultures evolve. Still little causal evidence has been collected. Here we deal with this dual challenge using a 30-day collective-risk social dilemma experiment to measure norm change in a controlled setting. We ask whether a looming risk of collective loss increases the strength of cooperative social norms that may avert it. We find that social norms predict cooperation, causally affect behavior, and that higher risk leads to stronger social norms that are more resistant to erosion when the risk changes. Taken together, our results demonstrate the causal effect of social norms in promoting cooperation and their role in making behavior resilient in the face of exogenous change.

[1] Institute of Cognitive Sciences and Technologies, Italian National Research Council, Rome, Italy. [2] Institute for Futures Studies, Stockholm, Sweden. [3] Collegio Carlo Alberto, Turin, Italy. [4] Grupo Interdisciplinar de Sistemas Complejos (GISC), Departamento de Matemáticas, Universidad Carlos III de Madrid, Leganés, Spain. [5] Institute for Research on Population and Social Policies, Italian, National Research Council, Rome, Italy. [6] Instituto de Biocomputación y Física de Sistemas Complejos (BIFI), Universidad de Zaragoza, Zaragoza, Spain. [7] Unidad Mixta Interdisciplinar de Comportamiento y Complejidad Social (UMICCS), UC3M-UV-UZ, Leganés, Spain. [8] UC3M-Santander Big Data Institute (IBiDat), Universidad Carlos III de Madrid, Getafe, Spain. [9] Malardalens University, Vasteras, Sweden. [10] These authors contributed equally: Aron Szekely, Francesca Lipari. ✉email: aron.szekely@carloalberto.org

From climate change and habitat destruction to the spread of infectious diseases, many contemporary societal challenges pose collective action problems where groups benefit from a shared outcome but individuals' incentives drive them to free ride on others' efforts. While laws and other formal institutions can foster cooperation to address global issues, they are often unavailable, unenforceable, or insufficient[1] and informal institutions like social norms become essential[2,3]. Despite their broad scientific[4–6] and practical importance[7–9], the factors that lead beneficial social norms to arise and change are insufficiently understood. According to the tightness-looseness theory of culture[6,10], in societies that experienced high threats (e.g., due to frequent disease, environmental catastrophes, or intergroup conflict) "tight" cultures emerge with stronger social norms and low tolerance of deviant behavior, thereby enhancing order and social coordination allowing these cultures to effectively manage the risks they face. Conversely, in low threat settings "loose" cultures arise that allow greater flexibility at the individual level but are less able to overcome risks at the collective one. Social norms, hence, are building blocks of culture that play a crucial role in shaping behavior. Importantly, causal evidence that social norms change in response to threat variants and that stronger norms increase social coordination is lacking. Most studies that test norm-change interventions on behavior, such as reducing antibiotic prescribing, discouraging binge drinking, and encouraging hand washing[11] do so without explicit measures of social norms and their strength. Other studies, which do measure social norms and their causal effect on behavior[8,12–14], adopt static approaches that cannot address norm change. As a consequence, social norms are both hailed as solutions[3] yet remain too vague for others[15].

To test whether social norms change in response to external threats, whether they causally motivate behavior, and how this affects their ability to solve cooperation problems we conducted a 30-day online experiment ($n = 286$) (Fig. 1). Compared to standard short-term designs, a longer-term experiment may allow norms to emerge and evolve. We use an extensive set of measures[16] to identify social norms, establish their causal effect

on behavior, and measure their change over time in the context of a modified collective-risk social dilemma[17,18]. Following Bicchieri[2], we define social norms as informal behavioral rules that individuals follow conditionally on their belief that: (i) a sufficiently large number of people in their community conform to the rule (empirical expectations), and (ii) a sufficiently large number of people in their community think that they ought to conform to the rule and may be willing to sanction transgressions (normative expectations). In this view, a social norm exists when there is a set of individuals who are disposed to follow a behavioral rule because they believe that both these conditions are fulfilled.

We elicit subjects' empirical expectations and normative expectations in each of the 28 game-days to detect the basic conditions for norms to motivate behavior. If social norms explain cooperative behavior, we expect that both empirical and normative expectations are positively associated with cooperation (Hypothesis 1). Crucially, we elicit empirical and normative expectations daily using distributions, not averages, allowing us to identify convergence of expectations accurately and detect multiple or conflicting norms. We also manipulate social expectations in a "conditional contribution" phase (i.e., with the strategy method) on a subset of the game rounds to identify the causal effects of empirical and normative expectations on behavior. We expect that the manipulation of empirical and normative expectations will result in changes in cooperation level such that higher empirical and normative expectations elicit higher contributions (Hypothesis 2). On the final day of the experiment after all contribution decisions have been made, we give subjects a one-time punishment opportunity. If social norms exist in our setting, we expect that subjects will target uncooperative subjects with punishment (Hypothesis 3a) and that subjects should also expect that uncooperative subjects are punished (Hypothesis 3b)[7]. We elicit punishment preferences only at the end of the experiment in order to avoid influencing the in-game behavioral dynamics and rule out instrumental motivations for punishment. To test whether different threats increase the strength of social norms, we manipulate the risk of the collective loss (high risk vs low risk)

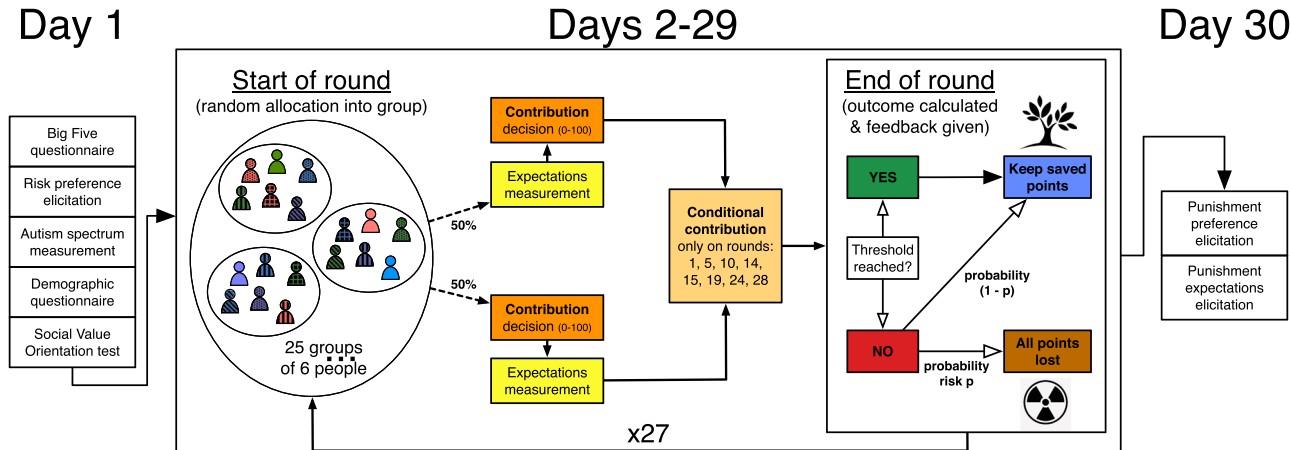

**Fig. 1 Structure of the experimental setup.** Each day represents an experimental round. During day 1, participants performed four individual trait tests: Big Five, risk preference elicitation, Autism spectrum and Social Value Orientation, and completed a short questionnaire. From day 2 to day 29 participants were matched in groups of six people and interacted according to the collective-risk social dilemma answering questions on their expectations (asked in a random order before or after the contribution decision) and deciding their actual contribution and their conditional contributions. Conditional contributions were elicited on rounds 1, 5, 10, 14, 15, 19, 24, and 28. At the end of each round, participants kept their saved points if the threshold was reached, otherwise they lost all their round points with probability $p$. The outcome of the conditional contribution decisions was calculated based on the contributions and actual empirical and normative expectations of each group (see Supplementary Information Section 4 for details). Each day the groups were reshuffled. During day 30, participants went through a punishment phase in which their punishment behavior and expectations were elicited. Subjects subsequently received information about their results and payment.

and we predict that different risk levels will influence the strength of the evolved social norms. Moreover, by modifying the order of the different risks, we examine whether stronger norms make the behavior more resistant to change than weaker norms. If so, we expect that the change in behavior is slower when a highly probable threat becomes less risky than when a low risk one becomes highly probable (Hypothesis 4)[10]. (see Methods for preregistered hypotheses and analysis plan). In this way, we provide an empirical account for how variation in experienced risk leads existing social norms to change and how this affects individuals' ability to solve collective action problems.

We also isolate the effect of social norms from individual-level factors that may determine contribution in our setting by measuring subjects' own beliefs about the appropriate way to behave (personal normative beliefs[2,16]) that are not conditional on the expectations of others, a range of dispositional traits that plausibly affect contribution and/or sensitivity to social norms (The Big Five[19,20], Social Value Orientation[21], Risk Preferences[22], Autism Spectrum Quotient[23,24]), and demographic variables (Supplementary Table 1).

In our collective-risk social dilemma, groups of six individuals can avoid the risk of collective loss (the threat) by spending money from their endowment (100 points every round) to reach a common threshold (300 points). If the threshold is reached the collective risk is averted and subjects keep their unspent points; otherwise, they risk losing their earnings for that round with probability $p$. The experiment was coded in oTree[25] (IBSEN version) and participants were recruited through the IBSEN subject pool (http://www.ibsen-h2020.eu) and participated in one of two experimental sessions involving 148 and 138 individuals each in June 2018 and September 2018, respectively (the Supplementary Information provides an English translation of the experimental instructions in Section 1 and information on participants' demographics, dropout rates and summary statistics in Section 2, Supplementary Table 1 and Supplementary Table 2). At the end of every round, subjects are informed about the contribution of their group members, the outcome, if a collective loss occurred, and their individual payoff for that round. At every new round/day, groups of six people were formed by shuffling active (non-excluded) individuals. Excluded individuals are those who either missed the first day or at least 4 decisions over the remaining 29 days. Reshuffling individuals among groups allows us to study how norms that regulate interaction among strangers emerge and change at the level of populations instead of within small fixed-groups embedded in long-term relations[26,27]. Moreover, reshuffling reduces issues from attrition since excluded subjects can be grouped together after every round. Since subjects do not know each other's identity, the expectations that we elicit from them do not concern behaviors or beliefs of specific others; instead, they concern generalized others' behavior and beliefs and thus demonstrate how general norms can emerge from small-group interactions.

To test how risk affects social norms, cooperation, and norm stability, we implement a within-subjects design. We manipulate the risk probability (high: 0.9 vs low: 0.6). Once the risk probability changes, it is stable for the remaining rounds; participants face one risk probability for rounds 1–14 and in the remaining 15–28 rounds they face a different risk probability. The between-subjects treatments vary the ordering in which subjects face different risks: in *High Low,* they first experience high risk (rounds 1–14) followed by low risk (rounds 15–28) while in *Low High* this order is reversed.

We evaluate the strength of social norms using three criteria: (i) agreement between expectations of group members (*consistency*[2]), (ii) whether expectations correctly predict the behavior and personal normative beliefs of others (*accuracy*[2]),

and (iii) how specific the norm is concerning the range of acceptable behaviors (*specificity*)[6] (see Section 5 of the Supplementary Information for precise definitions and the measurement scale; see Supplementary Section 3 for full model outputs of all analyses discussed below). Punishing intensity and expectations are elicited only in the last round of the game, hence by construction, our norm strength index does not include it. While the first two criteria capture the emerging group consensus around a social norm and are consistent with Bicchieri's conceptualization as well as other empirical studies on norm measurement[2,13,14], the criteria of specificity come from psychology and social science[6,19]. The strength of a norm is content free, meaning that strong norms are highly consistent, accurate, and impinge on a specific behavior regardless of the particular behavior they prescribe. To create a single norm strength measure, we define *norm strength = consistency × accuracy × specificity*, where each of the single factors are normalized to account for temporary differences in group size due to inactivity or exclusion (consistency, accuracy, and specificity turn out to be highly positively associated with each other: $r_{min} = 0.81$, $r_{max} = 0.99$).

## Results

**Social norms and contribution.** We start by testing the association between contribution and social expectations (empirical and normative expectations), personal beliefs (personal normative beliefs), and dispositions based on unconditional contributions and expectations. Social expectations are strongly and positively associated with contribution in all model specifications (Table 1, Supplementary Table 3) (Hypothesis 1). Every unit increase in empirical expectations and normative expectations is associated with 0.59 and 0.52 higher contributions respectively (Model 1) and together account for 22% of the variation in contributions. Personal normative beliefs also have strong and consistently positive significant associations with contributions. Including this variable increases the $R^2$ of the previous model to 31% (Model 2). In contrast, adding variables for dispositions and other factors only improves the model $R^2$ by an additional 4% (models 3 and 4). The results are robust with respect to attrition and the time-varying sessions. We manage attrition in two ways.

**Table 1 Predictors of contribution.**

| Independent variables | Dependent variable: contribution | | | |
| --- | --- | --- | --- | --- |
| | Model 1 | Model 2 | Model 3 | Model 4 |
| Empirical expectations | 0.590*** | 0.477*** | 0.479*** | 0.447*** |
| | (0.105) | (0.103) | (0.101) | (0.098) |
| Normative expectations | 0.521*** | 0.212** | 0.214** | 0.224** |
| | (0.116) | (0.081) | (0.079) | (0.076) |
| Personal beliefs | No | Yes | Yes | Yes |
| Preferences and psychological variables | No | No | Yes | Yes |
| Additional controls | No | No | No | Yes |
| Constant | −7.324 | −14.968** | −19.383** | −16.212* |
| | (6.613) | (5.149) | (7.802) | (7.847) |
| $R^2$ | 0.22 | 0.31 | 0.34 | 0.35 |
| Observations | 7433 | 7433 | 7433 | 7433 |

Linear regression model estimates with standard errors clustered according to 284 subjects. Statistical significance calculated using two-sided *t*-tests. Preferences and psychological variables contain social value orientation, risk preferences, Autism Spectrum Quotient, and a set of Big Five variables for extraversion, agreeableness, conscientiousness, neuroticism, and openness. Additional controls contain variables for treatment, collective risk, age, gender, student, experience with experiments, and left-right political orientation.
*$p < 0.05$, **$p < 0.01$, ***$p < 0.001$.

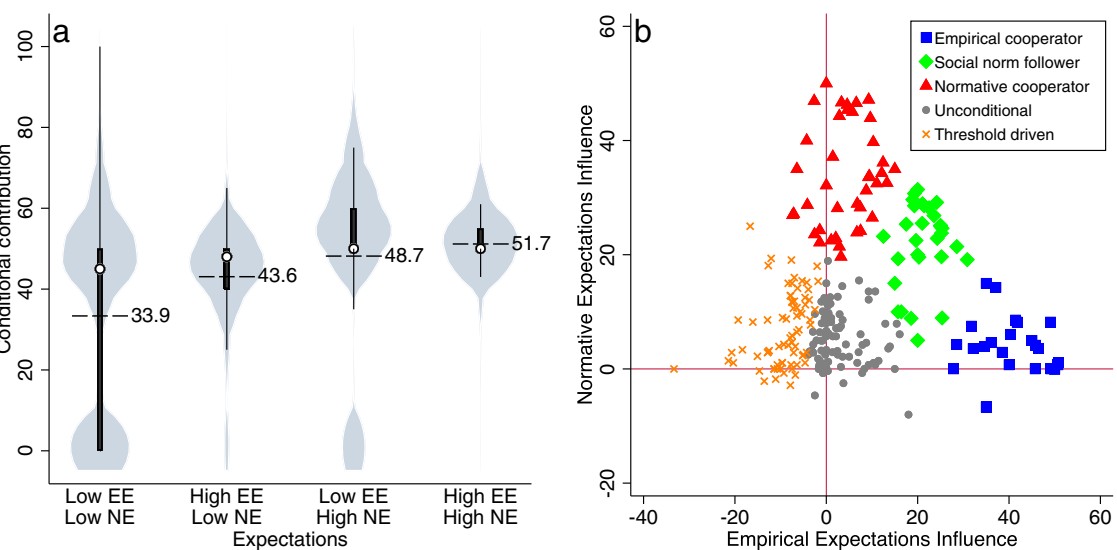

**Fig. 2 Conditional contributions and behavioral types. a** Conditional contribution according to empirical expectations (EE) and normative expectations (NE). High expectations imply at least 50 points and low expectations imply <50 points (e.g., the distribution for high normative expectations and low empirical expectations displays conditional contributions if the majority of one's group contributes <50 and believe that you all should contribute at least 50). $n = 283$ individuals repeatedly measured during the experiment. Dashed lines and text values show means, hollow circle markers display medians, bars represent the interquartile range, and spikes the upper- and lower-adjacent values (see also Supplementary Fig. 1). **b** The horizontal and vertical lines divide the space according to subjects' reactions with respect to a change in, respectively, normative and empirical expectations. Along those lines the change in contribution, due to a change in the corresponding expectation, is null. $n = 276$ individuals. Source data are provided as a Source Data file.

First, reshuffling players among groups, by design, allows us to account for attrition on group dynamics. Second, the sample composition from round 1 to round 28 remains constant throughout the experiment (Supplementary Table 2), and this suggests that subjects are randomly dropping out and attrition does not bias our results. We checked whether the starting sample characteristics across the two experimental sessions vary and find that they do not (Supplementary Table 1). Unconditional contributions and expectations covary in a feedback loop. Our next test demonstrates the causal effect of expectations on contribution.

We also find that contributions are causally and substantially influenced by empirical and normative expectations (Hypothesis 2). Figure 2a represents the subjects' conditional contributions in response to empirical and normative expectations. Mean contribution is 33.90, 95% CI [31.34, 36.47] in response to low empirical and low normative expectations, increases to 43.57, 95% CI [42.51, 44.64] under high empirical and low normative expectations, increases further to 48.74, 95% CI [46.42, 51.07] under low empirical and high normative expectations, and reaches 51.73, 95% CI [51.04, 52.43] when both empirical and normative expectations are high (Supplementary Table 4). Participants increase their contribution in response to higher empirical and normative expectations even reacting to a belief (normative expectation) that has no material implication for them. Strikingly, subjects respond more to normative expectations than they do to empirical expectations (difference, $p < 0.001$).

Further indicating the presence of cooperative social norms, low contributors (<50), irrespective of risk, are punished with a higher intensity (6.18, 95% CI [5.70, 6.67]) than those who contribute 50 (2.33, 95% CI [1.94, 2.72]) or more (2.02, 95% CI [1.61, 2.44] points (Hypothesis 3a) and subjects expect low contributors to be punished with a higher intensity (7.03, 95% CI [6.62, 7.46]) than higher contributors (contribute 50: 3.37, 95% CI [2.93, 3.80]; contribute more than 50: 2.82, 95% CI [2.35, 3.28]) (Hypothesis 3b) (Tables S5 and S6).

**Behavioral types based on social expectations**. To understand why cooperation changes according to expectations, we use $k$-means clustering[28] to generate a stringent classification of subjects (Fig. 2b) based on their responsiveness to empirical and normative expectations (see Supplementary Information Section 6; Supplementary Fig. 2, Fig. 3; Supplementary Table 7 for individual predictors of responsiveness). Social norms followers often manifest themselves as conditional cooperator types[29] yet the motivation for conditional behavior remains unclear. Consistent with extensive literature[29,30], we find conditional cooperators but we move beyond this typology, which is based on behavior, and identify the types of expectations that shape their conditional cooperation. Our method further identifies three sub-types of conditional cooperators whose behavior depends on different response to expectations (consistent with Bicchieri's work[2,7]): *empirical cooperators* (11.6%; 32/276 of the participants), who cooperate primarily because they think others will also cooperate (empirical expectations), *normative cooperators* (14.1%; 39/276) who cooperate primarily because they think others think that they ought to cooperate (normative expectations), and *social norm followers* (10.9%; 30/276) who cooperate due to both empirical and normative expectations, that is put a similar weight on both. We also find *threshold-driven participants* (26.5%; 73/276) who decrease contribution when others increase[31] and *unconditional* participants (37%; 102/276), who do not change their behavior in response to expectations. If we use a less stringent notion of social norm influence based only on a positive response of contributions due to normative expectations we find that 77.2% (213/276) of our subjects are positively influenced to some extent, increasing contribution by >1 in response to high normative expectations.

**Social norm strength in risky environments**. We now turn to the between treatment comparisons. In line with the conjecture that tighter norms emerge in higher risk environments[6,10], we find stronger social norms in the high-risk settings (Fig. 3, Supplementary Fig. 4). Norm strength is 0.80, 95% CI [0.79, 0.81] in high risk settings and 0.72, 95% CI [0.71, 0.73] in low risk settings

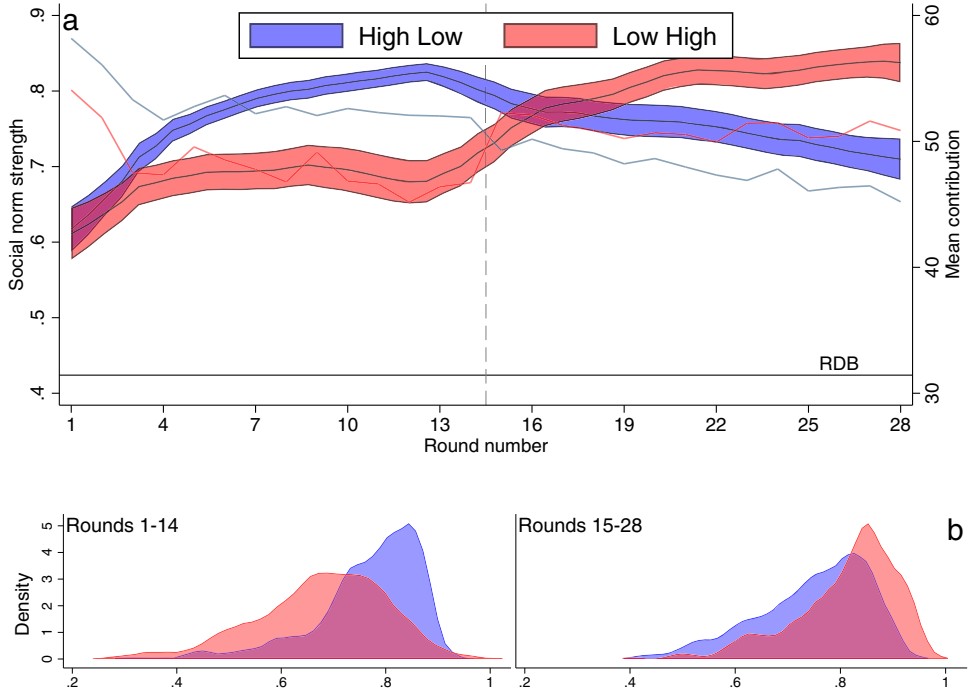

**Fig. 3 Social norm and contribution dynamics. a** Norm strength (consistency × accuracy × specificity) as a function of the round. RDB indicates the Random Decision Baseline of 0.424 (mean identified using 1000 simulations); the norm strength that would be observed if subjects made their decisions randomly for contribution, empirical expectation, normative expectations, and personal normative beliefs. Shaded areas indicate 95% CIs with one observation per group; based on $n = 284$ individuals. Solid lines without shading indicate mean contributions (blue: High Low; red: Low High). **b** Distribution of norm strengths in the two stages of the experiment at the group level (one observation per group based on $n = 284$ individuals). Source data are provided as a Source Data file.

(difference: $p < 0.001$, Cohen's $d = 0.772$) (Supplementary Table 8). To help interpret these values, we used simulations to calculate the social norm strength in a Random Decision Baseline at 0.424. This is the social norm strength that would be achieved by individuals making random decisions for their contribution, empirical expectation, normative expectations, and personal normative beliefs. The differences in social norm strength translate into substantial differences in average cooperation (high risk: 52.18, 95% CI [51.39, 52.97]; low risk: 48.00, 95% CI [46.47, 49.53; Supplementary Table 9) and the ability of groups to reach the threshold: 0.48, 95% CI [0.44, 0.52] proportion do so under low risk and 0.75, 95% CIs [0.72, 0.79] achieve it under high risk (Supplementary Table 10). Moreover, groups with stronger social norms are likelier to reach the threshold level than groups with weaker social norms (OR = 5.793, s.e. = 3.049, $p = 0.001$) (Supplementary Table 11, Supplementary Fig. 6).

As norms are stronger and contributions higher in high risk, we expected cooperation to prevail in the 0.9 part of High Low and to slowly break down following a change in risk. On the other hand, we predicted that cooperation is less established in the 0.6 part of Low High but will emerge after the change in risk. We test this by comparing average contributions in the two treatments at round 15 relative to what they were in the other treatment in round 14 and analyzing the change of contributions from round 15 to round 28. We use round 14 contributions in the other treatment as a baseline comparison (i.e., round 14 in High Low against round 15 Low High and round 14 in Low High against round 15 High Low) because these are the contribution levels that are observed for the same risk level; for instance, round 14 in High Low has risk 0.9 and round 15 in Low High has risk 0.9.

Consistent with this hypothesis (Hypothesis 4), in round 15, average contribution in Low High (52.26) is already comparable to the average contribution of subjects in round 14 of High Low

(51.89; difference: $b = 0.368$, $p = 0.738$) and this contribution level remains stable until the end of the experiment ($b = -0.061$, $p = 0.216$) (Fig. 3). In contrast, contribution in High Low decays slowly when the risk decreases. In round 15 of High Low, contribution remains higher (49.29) than the round 14 contribution in Low High (46.72; difference: $b = 2.568$, $p = 0.012$) and this decays as the experiment proceeds ($b = -0.331$, $p = 0.001$) (Supplementary Tables 12, 13). The difference between the two treatments in their round 15 proximity to their baseline comparisons is near significant although with large uncertainty ($b = 2.937$, $p = 0.051$, 95% CI [−0.01, 5.88]), while the differences between the treatments in the trajectory of contributions over rounds 15–28 are significant ($b = 0.270$, $p = 0.017$, 95% CI [0.05, 0.49]). In addition, in the high part of Low High the percentage of groups reaching the threshold 0.67, 95% CI [0.62, 0.73] is significantly lower than in the high part of High Low 0.82, 95% CI [0.78, 0.86], further demonstrating the inertia effect that social norms can have on behavior (Supplementary Table 10).

Consistent with stronger social norms in the high-risk treatment, punishment of low contributors is high and marginally higher when the rounds directly before punishing elicitation are high risk, at 6.67 points, 95% CI [5.98, 7.36], than when these rounds are low risk at 5.78, 95% CI [5.11, 6.46] (difference: $p = 0.071$) (Supplementary Table 5). Moreover, subjects expect that punishing of low contributors is more intense in the high-risk setting. Their expectation in low risk is 6.64, 95% CI [6.05, 7.22] and in high risk are 7.52, 95% CI [6.93, 8.11] (difference: $b = 0.88$, $p = 0.039$) (Supplementary Table 6). On the other hand, punishing towards contributors, 50 or above, does not differ depending on the risk of the prior rounds. Punishing towards 50 contributors is 2.44, 95% CI [1.90, 2.97] in low risk and 2.20, 95% CI [1.63, 2.78] in high risk, whereas it is 1.99, 95% CI [1.42, 2.55] in low risk and 2.07, 95% CI [1.45, 2.69] in high risk towards those who contribute more than 50

(Supplementary Table 5). Punishing expectations of contributors reflect the same finding (Supplementary Table 6).

## Discussion

Consistent with our hypotheses, we find greater cooperation and stronger social norms in the high-risk environment and slower behavior change after a change in risk when social norms are stronger. Moreover, social norms predict cooperation, causally affect behavior, and lead to the punishment of norm-breakers. Our results show that high risk of collective loss increases social norm strength, reduces tolerance of deviance, and increases cooperation. Risk changes norms and coordinates and motivates social expectations and contributions leading to higher cooperation under high risk, whereas stronger norms provide greater resistance to behavioral change than loose ones. This is consistent with the tight-loose theory of cultural change and is, to our knowledge, the first causal evidence in support of this key assumption[6,10]: exposure to threats leads to the emergence of stronger norms for organizing social interaction. Yet, our results indicate that when the risk of a future threat decreases, norm strength may diminish as well: an issue that has important implications when designing policy interventions. Thus, once the social dilemma has been provisionally averted, it may be important to establish additional mechanisms to support compliance with the evolved social norms and to help sustain cooperation even further into the future. One possibility, along with others such as communication, is to favor the adoption of a distributed enforcement mechanism like peer punishment. Somewhat counter-intuitively, our results suggest that the appeal to enforcement mechanisms may be especially important when the risk of collective loss is low because social norms start losing their strength. We also demonstrate that social expectations are stronger drivers of cooperation than personal dispositions and individual factors that we measure, in line with previous work[32]. This suggests that it may not be necessary to shape individuals' dispositions in order to change behavior: shaping expectations, which is often easier to do, can be sufficient[32,33]. Another key observation is that social norms causally motivate cooperative behavior and different individuals respond differently to social expectations. Empirical cooperators respond positively to what they expect others to do while threshold-driven respond negatively to their empirical expectations; normative expectation-driven cooperators and social norm followers respond positively to an increase in normative expectations. This implies policy making must embrace heterogeneity, designing interventions that reach all types based on both empirical and normative expectations and avoiding potentially harmful messages that focus only on empirical affairs that may backfire for some individuals (threshold types in our context)[34]. Finally, we have shown that a precise, measurable definition of social norm allows quantitative analyses lead to insights about the feedbacks between norms and behavior, leading in turn to specific predictions of optimal behavioral change interventions relevant to global societal challenges. Yet extensive space for research remains on the role of social norms in social dilemmas. One particularly important question is what would happen if the condition of threat is even less extreme (e.g., the probability of collective loss is <0.5 or the amount lost if it does occur is little) such that groups would be better off in expectation from individuals not contributing. It might then be possible that "harmful" social norms supporting cooperation emerge, leading to accelerated detrimental effects. Further research along these lines is urgently needed.

## Methods

**Experiment**. We designed a 30-day online experiment using oTree[25]. Subjects, on the first day, completed the Big Five personality questionnaire, the Social Value Orientation slider measure, the Autism Spectrum questionnaire, and a risk preference elicitation task. Subsequently, they participated in 28 rounds (one round

per day) of our variant of the collective-risk social dilemma. At the end of the experiment (day 30), we elicited subjects punishing behavior, their expectations concerning punishing, and they answered a short end questionnaire. All the interactions in the experiment took place anonymously on computers or phones.

In the daily collective-risk social dilemma, subjects are randomly allocated into groups of 6 and each subject receives 100 points endowment at the start of every round, and decide how much of this to contribute to the collective pot. All contributions to the pot are destroyed. If a threshold amount (300 in total) is reached, subjects keep what they did not contribute as they avert the collective loss. If the threshold is not reached, there is a probability ($p$) with which subjects have all of their remaining endowment from that round destroyed and a probability (1-$p$) with which they keep the points that they did not contribute. Contribution decisions were made simultaneously and subjects were not able to communicate with one another. Subjects were informed after each round of the contributions of other group members. There were no identifiers for other players and the ordering of contributions in a group was randomly determined. Groups were reshuffled every round. Subjects were either "active", "inactive", or "excluded". Active subjects are those that completed all of their decisions on a given day, inactive subjects are those who are still actively participating in the experiment (i.e., did not miss day 1 and have not missed 3 or more decisions in-game) but missed some decisions. Excluded subjects are those who have been removed from the experiment because they missed their decisions on day 1 or missed 4 or more decisions during the game. To reduce the effect of inactive and excluded participants on our experiment, excluded subjects were grouped together after each round, and inactive and excluded subjects made their decisions for contributions, empirical expectations, normative expectations, and personal normative beliefs, by randomly selecting a value from the other active participants in their group.

We implement four treatments in a mixed within- and between-subjects design. The within-subjects treatments change the risk probability (0.9 or 0.6). Subjects face one risk probability for 14 rounds and in the other 14 rounds, they face a different risk probability. The between-subjects treatments vary the ordering: whether subjects face a 0.9 risk and then a 0.6 risk or vice versa.

To study whether subjects' contribution was driven by the presence of social norms, we elicited subjects' Personal Normative Beliefs (PNB), Empirical Expectations (EE), and Normative Expectations (NE). We elicited these beliefs every round and randomly before or after their contribution decisions. Both the EE and NE elicitations were incentivized by comparing the former against contribution of their group and by comparing the latter against the PNB of their group (see Section 4 of the Supplementary Information for further details). In addition, on selected rounds of the collective-risk social dilemma (1, 5, 10, 14, 15, 19, 24, 28), we asked subjects to report their conditional contributions. This asked subjects how much they will contribute to four combinations of high and low EE and NE: if the majority of their group members put in [at least 50 points/<50 points] and believe that you should all spend [at least 50 points/<50 points]. We incentivized this by identifying the EE and NE combination that held in the subject's group and they were additionally paid for this.

Subjects, 286 of whom participate in the experiment, were recruited from the IBSEN subject pool (http://www.ibsen-h2020.eu). They are Spanish or residents in Spain, are 30 years old on average, 56% are female, and 50% of them are students. They earned an average of €19.97 ($23.62) including a €5 show-up fee. To reduce dropout, one randomly selected subject received a 10* multiplier to their earnings (to a maximum of €200). The research sample followed the requirements of representativeness for lab/online experiments. Participants were contacted through email and they played the on-line experiment either via computer or via mobile phone. The data collection occurred in two sessions that lasted 30 days each: the High Low session started on 4 June 2018 while the Low High session started on 3 September 2018. A total of 23 subjects dropped out or were excluded during the experiment (3 in High Low and 20 in Low High).

We preregistered our study hypotheses and analysis plan on the Open Science Framework at https://osf.io/f3cyt[35].

The study complied with all relevant ethical regulations for work with human participants. Informed consent was obtained from all subjects. The study received institutional ethical approval from the Institute of Cognitive Sciences and Technologies ethics board (Italian National Research Council, Rome, Italy).

**Statistical methods**. The statistics presented in the paper use parametric tests. We used different regression models (OLS with clustered errors, logistic regression). Section 3 in the Supplementary Information provides information on the different models used to test the hypotheses of the paper. Data were analyzed using Stata IC 16.1 and R 3.6.3. The decisions of inactive and excluded subjects were removed from the data before analysis.

**Reporting summary**. Further information on research design is available in the Nature Research Reporting Summary linked to this article.

## Data availability

The data generated in this study have been deposited in the Open Science Framework (https://doi.org/10.17605/osf.io/wvgk9)[35]. Source data are provided with this paper.

## Code availability

The experiment and analysis code are available at the Open Science Framework (https://doi.org/10.17605/osf.io/wvgk9)[35].

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

## Acknowledgements

This research was funded by the Knut and Wallenberg Grant "How do human norms form and change?" 2016.0167 awarded to G.A.; A. Sánchez acknowledges partial support by PGC2018-098186-B-I00 (BASIC, FEDER/MICINN- AEI), PRACTICO-CM, (Comunidad de Madrid), and CAVTIONS-CM-UC3M (Comunidad de Madrid/Universidad Carlos III de Madrid); M.P. acknowledges funding by the FLAG-ERA JCT 2016 FuturICT2.0 project; A.A. acknowledges partial support from the Ministerio de Economía y Competitividad of Spain Grant No. FJCI-2016-28276; F.L. acknowledges partial support from the Comunitad de Madrid Talento Scholarship Grant No. 2018-T2/SOC-11335. We thank Antonello Maruotti and Guillaume Kon Kam King for scientific discussions and valuable feedback.

## Author contributions

A. Szekely, F.L., A.A., M.P., A. Sánchez, L.T., and G.A. designed research, A. Szekely and A.A. wrote the experimental code, A. Szekely, A.A., M.P., and F.L., performed the experiments, A.Szekely, F.L., and A.A., analyzed the data and performed the statistical analysis, A. Szekely, F.L., A.A., M.P., A. Sánchez, L.T., and G.A. prepared the initial version of the manuscript, A. Szekely, F.L., A.A., M.P., A. Sánchez, L.T., and G.A. revised and approved the final version of the manuscript.

## Competing interests

The authors declare no competing interests.
