## [Peer Review File · Nature Communications]

Evidence from a long-term experiment that collective risks change social norms and promote cooperationREVIEWER COMMENTS

Reviewer #1 (Remarks to the Author):

Review of Collective Risks Change Social Norms and Promote Cooperation.

This is a very important paper and makes a number of contributions. Methodologically, the authors advance the measurement of the strength of norms. Theoretically and empirically, the authors provide a very interesting causal test across time of the role of collective threat and cooperation. The asymmetry that is found is also very interesting, particularly the cultural inertia observed in the High-Low condition. I have a few questions and suggestions below to further strengthen the paper.

1. The theory and hypotheses needs to be better articulated. Currently the reader is referred to a pre-registration for the hypotheses but they should be in the main text. What is the rationale, for example, for why you expect slower behavior change after a change in risk when social norms are stronger? Is there any evidence that would support this very interesting asymmetry?

2. The components of the social norm strength measure are clearly articulated, but it is unclear why punishment expectations (which was only measured at the very end of the experiment) wouldn't theoretically be part of a measure of norm strength. For example, I might think others will cooperate and think I should cooperate, but I may expect no punishment for not cooperating, which would presumably affect my own decisions on whether to cooperate or defect. At the very least, I would suggest that you discuss this as a limitation/ future direction for future research on the topic. One could build another measure of punishment strength that has similar components to the norm strength component (e.g., including accuracy, consistency, and specificity) and it may explain additional variance in cooperation (particularly because there is a lot of unexplained variance).

3. Related to the above, the label of "specificity" is a bit unclear. Perhaps "range of variation" be a better label.

4. The manipulation of threat was arguably about the high vs. low probability of extreme threat. What would you expect when in other conditions of threat that are less extreme? This could go in the discussion section.

5. The role of the individual traits in the study is unclear. Why were these individual differences chosen? Did any of them interact with the treatment? I'd expect that individuals high in risk aversion would show even more cultural inertia going from High to Low for example. Or likewise, does the relationships between NE and EE vary depending on the moderators. Overall, the theorizing and empirics on these traits is a bit weak.

6. Related to the above, it would be useful to have a table that shows the individual traits in the main models (Table 1 just lists them collectively so isn't really informative for the reader).

7. For Table S8, does strength of norms predict above and beyond the controls and traits. I'd image it would but they aren't in the model.

8. P 7. Can you motivate the cluster analysis more from a theoretical point of view, and also specify what we learn from this analysis. It seemed disconnected from the main research question of the HL and LH treatment. Were the same proportion of cluster found in each of these treatments? Perhaps the HL produced a higher percentage of the social norms followers, for example.

9. As an aside, I found the label "social norm followers" a bit awkward b/c the other strategies were also norm followers.

10. Page 12 of the supplemental. Do you mean accuracy rather than consistency in this discussion?

Reviewer #2 (Remarks to the Author):

This is an interesting paper that tests important questions about cooperation, norms, and norm changes. Among several key results, they find that stronger norms emerge when people face a greater risk (compared to lower risk) in a “collective risk social dilemma” and that if strong norms are established first they change only slowly, but if weak norms are established first, they change quickly. Although many of the main results have to do with correlational relationships between norms and cooperation, the article also uses an interesting approach to get a causal handle on how changes in norms could affect cooperation (the “conditional cooperation” decisions).

I think this paper could potentially be publishable in an outlet like NC, but I have several questions, some serious, about the design and analysis. Although some cut to the heart of the paper, it's possible the authors could deal with them all in a satisfactory way.

To me, the most striking finding is one that actually seems to work against the authors' conclusions: their own figure 3b and associated text. Roughly 75% (!) of the participants more or less ignore their own normative expectations when deciding how much to contribute. This seems bad for an account based on norms. The single largest subsample, 37% are impervious to both normative and empirical expectations. 38% of players based their behavior in some way on their empirical expectations – and only empirical expectations (some contributing more when others do, some trying to contribute just enough as a group to meet the threshold). Broadly, this latter set is consistent with cooperation in a coordination game (which the present game is), regardless of whether there are norms involved. Perhaps I am misunderstanding something, but the authors would need a convincing explanation for why these results don't actually work against their view of the importance of norms. There are many non-normative reasons empirical expectations would matter in a coordination game.

Related, Figure 3b needs to be re-drawn. The authors visually privilege their hypothesis by stretching the normative axis relative to the empirical axis. (I.e., it takes twice as much change on the empirical axis to look like as big as an identical change as on the normative axis.)

Theoretically, why are groups reshuffled every round? Typical discussions of norms most centrally focus on norm developing within a group. But the current design is more like a norm developing within a meta-population. I'm not sure if this is a problem but would like to know the authors' thoughts.

I do not believe the authors do a direct test of one of their main hypotheses: that the rate at which norms change is faster in low-high than in high-low. They show that there is a change in norms when there is a change in risk in low-high and they show that there is no change in norms when there is a change in risk in high-low. But they do not do the direct “difference in difference” test.

Norm strength is an incredibly important variable in the paper, but readers aren't given an intuitive sense of what a particular value means. Norm strength is .71 in the low risk condition. Is there anyway to interpret that absolute number as meaningful or is the metric purely arbitrary? This is relevant because the high risk condition had norm strength of .78. The difference was significant, but given how much power this study had (given a reasonable sample size and many, many repeated observations), it might be more relevant to see the effect size of the difference, either in a standardized metric like Cohen's d or some sort of interpretation that makes the raw difference of .07 substantively meaningful. The authors point to the differing success of the groups as relevant to showing the substantive importance of the difference in norm strength. But I think raw success is largely beside the point. If the risk level was 50% probability of disaster, risk-neutral players would be indifferent between contributing to meet the threshold and not bothering at all. The actual 60% risk in the low risk condition isn't much bigger than 50%. So there may be many players who, through some combination of low risk aversion and low confidence others will contribute enough, don't bother to contribute. It might have nothing to do with norm strength.

I was confused by a lot of the specifics involved in the norm strength measure and the component parts that make it up.

- 1) Most generally, it seems to hide a lot of information to simply multiply together the components of consistency, accuracy, and specificity. Shouldn't those be analyzed separately?
- 2) Related, multiplying measures together to arrive at a final DV to analyze is usually considered a poor approach because multiplying random variables radically increases the standard deviation of the resulting variable. On the other hand, the CIs for norm strength by condition show that these variables are surprisingly well estimated
- 3) Many issues were unclear in the computation of consistency, accuracy, and specificity. Each computation is supposed to compute for a specific player, p , their score. So why is there a summation for p in the formula? Either it's just cluttering the formula and represents a "summation" of 1 element or something else is wrong or confusing (to me, at least).
- 4) Similarly, why distinguish between group g and time t ? Participants are in a new group every time period, so aren't time and group effectively synonymous?
- 5) Is there any reason besides simplicity that EE and NE are weighted identically?
- 6) More generally, why are empirical and normative elements simply added together? They are different concepts. If a researcher measured personality traits of agreeableness and extraversion, it does not seem to be coherent to add those (raw) scores together and say you've created a general 'personality' score. (Perhaps you could standardize each score first and then aggregate for a "positive valence" personality score.) Are empirical and normative expectations etc different?
- 7) That that R subscript symbolizes rank number for a particular (e.g.) expected contribution is not formally stated; readers are left to infer it.
- 8) It's never explained why the formulas include a normalizing max term in the denominator.
- 9) Pairing things up for deviances by ranking two separate lists seems potentially arbitrary. Consider a person whose empirical estimations were (90, 80, 70, 60, 50) and the actual other group members contributed (100, 90, 80, 70, 60). Pairing up by ranks creates a uniform set of deviances of 10, yet just looking at the numbers suggest the person was quite accurate for 4 out of 5 group members, but they radically misestimated a single player, the player who actually contributed only 50. On the other hand, I played around with some sets like these and got the same set of absolute deviances regardless of whether I computed, taking this example, (90, 80, 70, 60, 50) - (100, 90, 80, 70, 60) or (90, 80, 70, 60, 50) - (90, 80, 70, 60, 100). In both cases the sum of the variances is 50. However, the latter version had greater variance in its deviances. So I'm not sure this is actually a problem but the authors should say more on why this is the appropriate strategy.
- 10) A similar question arises in the specificity measure. Why are participants' *individual* estimates for other players compared to the group average? Why not the participants' average estimates to the group average? In part this measure, as computed, could just be measuring attentiveness or thoughtfulness. Consider two players. Both have identical, perfect estimates of the average mind of their other group members. But one is lazy and just puts this average estimate down for all 5 group members. The other one thinks hard and tries to come up with a reasonable distribution around the average for the other group members. The lazy person will have perfect specificity merely because they are lazy. Perhaps this is an extreme case, but nonetheless more needs to be said about the specific measure's computational form.

Reviewer #3 (Remarks to the Author):

Please find attached.

[Reviewer Attachment is on the following page of the Peer Review File]

I read the paper with interest. However, in my view the paper over-claims what it can deliver and remains rather vague on aspects that are relevant to obtaining sufficiently important and externally valid insights. I provide helpful pointers below:

- The motivation and literature discussion seem disconnected from what the experiment does: (1) formation of cultures is a long process, the dynamics of which cannot be captured in a 30-day time window. (2) those changes have been the result of one-off exogenous shocks, whereas we are still in the midst of the pandemic. At the least, the study would need to examine post-pandemic behavior to make a strong claim about persistent changes in social norms.
- Relatedly, the authors emphasize their ability to study behavior in the context of a disaster, but in reality, the authors implemented a repeated threshold Public Goods Game with framing. To make broader, externally valid claims, you'd need to follow the existing research on natural disasters to capitalize on the impact of actual disasters with big implications for society.
- In terms of design, there is too much going on (Fig. 1 is everything but intuitive) and the experiment description is rather confusing. For example, the authors state that participants participated in two experimental sessions in June and September 2018. It took me a while to realize that this is not an indicating of a within-design (participants doing one experiment in June and coming back in September), but rather an indication that the authors collected data in two waves with different participants.
 - On this note, however, a lot of important information is missing: what about attrition over the course of 30 days and how was this accounted for in the analyses (e.g., one can imagine that certain types of people drop out along the way and thus introduce noise into the results)? Since data was collected with several months apart but still pooled in the analyses, it would be important to show that these participants do not differ along relevant dimensions.
- More generally, the extent to which the experimental design (reshuffling groups after each period) is consistent with the theoretical framework that the authors put forward (social norms are properties of reference networks) is unclear. If you want to study how norms change and develop, why would you want to shuffle the groups every time? Asking for beliefs at the 'global' rather than specific group level does not resolve this tension.
- I had a hard time following the authors' arguments about the chain of causality in their repeated game: both norm-related beliefs and actual behavior (with feedback!) are essentially elicited simultaneously. After day 1, the results are largely affected by endogeneity. Because feedback is given, participants are updating their beliefs about the environment, even if they are randomly matched with other participants next time (although no perfect stranger matching is attainable given the group sizes and number of

periods). The authors need to be more explicit about the causal inferences that they believe can be drawn from this design.

- The analysis was also rather confusing at times: for example, the pre-registration is powered for an alpha of 0.05, so it seems rather odd to refer to p-values above that as significant (e.g., line 140ff). On the other hand, Figure 3 illustrates a breakdown by low/high EE/NE, but I was unable to find a precise definition of what constitutes low and high. Without it, this classification appears rather arbitrary. Lastly, in contrast to the existing social preference literature that also includes classifications of free-riders, unconditional cooperators etc., the authors find 'new' types. It remains unclear what one can learn from this new classification as now distinct insights or policy implications are provided at any point.

REVIEWER 1

#	Question/Comment	Response
1	The theory and hypotheses needs to be better articulated. Currently the reader is referred to a pre-registration for the hypotheses but they should be in the main text. What is the rationale, for example, for why you expect slower behavior change after a change in risk when social norms are stronger? Is there any evidence that would support this very interesting asymmetry?	We have specified the hypotheses clearly and included them throughout our manuscript (p. 2, ll. 64-66, 71-83 and pp. 5-7) and we have included a citation suggesting such an asymmetry ¹ . We have also clarified our definition of social norm in the manuscript (p. 2, ll. 57, 60-62) and in Section 5 of the SI.
2	The components of the social norm strength measure are clearly articulated, but it is unclear why punishment expectations (which was only measured at the very end of the experiment) wouldn't theoretically be part of a measure of norm strength. For example, I might think others will cooperate and think I should cooperate, but I may expect no punishment for not cooperating, which would presumably affect my own decisions on whether to cooperate or defect. At the very least, I would suggest that you discuss this as a limitation/ future direction for future research on the topic. One could build another measure of punishment strength that has similar components to the norm strength component (e.g., including accuracy, consistency, and specificity) and it may explain additional variance in cooperation (particularly because there is a lot of unexplained variance).	We agree with the reviewer that punishment intensity and expectations are important factors contributing to the strength of a social norm. However, in our design punishment is measured only once at the end precluding the possibility of creating a dynamic measure across period (and if we add punishment only to the final day's strength index then it can longer be compared across rounds). We deliberately decided to adopt this design, with punishment only at the end, because including it before in every round would fundamentally change the structure of this situation. Our concern here is about social norm change in a "basic" setting without additional complexities such as punishment. Nevertheless, we agree that an important future step will be to include a punishment stage after every round in this setting thereby allowing us to also create a round by round punishment strength measure. We now mention the above points in the text (p. 2, ll. 73-77; p. 3, ll. 105-106; p. 4, ll. 146-148; p. 9, ll. 301-305).
3	Related to the above, the label of "specificity" is a bit unclear. Perhaps "range of variation" be a better label.	We have carefully reconsidered this label and ultimately prefer to keep "specificity", which, in our context means "the quality of being specific". We believe that this is consistent with what the measure aims to capture. When the distribution of permissible behaviours is wide, specificity is low (so the social norm is not specific or imprecise).

		Conversely, when the distribution of permissible behaviours is narrow, specificity is high (so the social norm is specific or precise). We explain this more clearly in Section 5 of the SI. We thank the reviewer for the suggestion of “range of variation”. Using this label however, would, in our view, make our index harder to understand because the meaning for the norm strength components would be in opposite directions: a higher value for range variation implies weaker norm strength while higher values for consistency and accuracy imply stronger social norms. More generally, we have clarified that the constructs of consistency and accuracy arise directly from the implications of Bicchieri’s theory of social norms² (p. 4, ll. 142-144) while specificity is derived from psychology and social science more generally (where it is sometimes called situational constraint)^{3,4}.
4	The manipulation of threat was arguably about the high vs. low probability of extreme threat. What would you expect when in other conditions of threat that are less extreme? This could go in the discussion section.	This is a very interesting point. It is not clear to us what would happen. Consequently, and as suggested by the reviewer, we include the reviewer’s point as a question for future research at the end of our article (p. 9, ll. 320-326).
5	The role of the individual traits in the study is unclear. Why were these individual differences chosen? Did any of them interact with the treatment? I’d expect that individuals high in risk aversion would show even more cultural inertia going from High to Low for example. Or likewise, does the relationships between NE and EE vary depending on the moderators. Overall, the theorizing and empirics on these traits is a bit weak.	Our primary motivation for measuring individual traits was to control for these factors and better capture the effect of social norms. We mention this and why these individual factors, specifically, were chosen (pp. 2-3, ll. 87-93). At the same time, we agree with the reviewer’s deeper point that the role of individual traits is a fascinating one. So we have conducted some additional exploratory analyses to test the reviewer’s point about risk (see Supplementary Information, Table S15). We do not find evidence that risk preferences affect inertia in contributions, but, we do find that personal normative beliefs do play a role in both treatments (the only variable to do so). After the change in

		risk, they prevent contributions from dropping in High Low and “push” them up in Low High. We are not entirely sure what relationship between NE and EE the reviewer is referring to since we did not analyse the relationship between the two. We think that the reviewer may have been referring to the individual trait predictors of Empirical Expectations Influence and Normative Expectations Influence. We have conducted additional analyses on this and include them in the Supplementary Information (Table S16) and we direct interested readers to the relevant analyses in the manuscript (p. 8, l. 268).
6	Related to the above, it would be useful to have a table that shows the individual traits in the main models (Table 1 just lists them collectively so isn’t really informative for the reader.	We mention the individual trait variables in the notes of Table 1 (p. 7, ll. 237-240) and we now refer interested readers to the SI, Table S1.
7	For Table S8, does strength of norms predict above and beyond the controls and traits. I’d image it would but they aren’t in the model.	It is important to clarify that in, previously numbered, Table S8 norm strength is the dependent variable (and not an independent variable). It is for this reason that we are not entirely certain which analysis the reviewer has in mind. Having said that, we think that the reviewer has in mind to predict contributions using social norm strength. If so, we believe that there is no theoretical link between social norm strength and level of contribution. One may have strong social norms about contributing little, weak ones about contributing a lot, or any combination. More generally, strong social norms may emerge about both prosocial behaviours (e.g. cleaning up litter) or harmful social norms (e.g. binge drinking). We intentionally designed our social norm strength index to be “content free” and have the flexibility to encompass the full range of combinations. We clarified this on p. 4, ll. 145-146. Nevertheless, we did conduct such an analysis to test the association between social norm strength and contributions using a linear regression. Here total group contribution is the dependent variable and social norm strength and risk are the

		independent variables. We find no association between contribution and social norm strength ($b=5.054$, $s.e.=8.747$, $p=0.563$; see Table S10). Moreover, a plot of the association shows no visible association, although it demonstrates that variation is lower at high social norm strengths (see Fig. S4). This suggests that strong social norms coordinate action at a particular level (consistent with our claim that a greater proportion of groups reach the threshold when social norms are stronger).
8	P 7. Can you motivate the cluster analysis more from a theoretical point of view, and also specify what we learn from this analysis. It seemed disconnected from the main research question of the HL and LH treatment. Were the same proportion of cluster found in each of these treatments? Perhaps the HL produced a higher percentage of the social norms followers, for example.	The main rationale behind the cluster analysis is that it is essential to identify heterogeneity with respect to individuals' preferences for compliance with social norms. This contributes to a more complete understanding of our experimental results. Moreover, we believe that the output of this analysis has important implication for the debate on the role of "conditional cooperators" who are posited to be motivated by social norms in the economic, psychological and sociological literature⁵⁻⁷. Yet, since this literature uses only behaviour to infer motivations, the tie with social norm is unclear. Here we use our detailed measures to tackle this challenge directly and obtain a detailed picture of the motivations of conditional cooperators. We find that the label "conditional cooperator" actually hides multiple sub-types that are motivated by different kinds of expectations, and, that only some of these can be classified as social norm followers in a strict sense. We have tried to make this clear in the section on types (p. 8, ll. 269-272) We have checked whether the same proportion of types were found in each treatment and generally do not find substantive differences. We have included a figure of the types broken down by treatment in the Supplementary Information (Fig. S6, Fig. S7). We have also added a second method (Type 2) for classifying behavioural types and find no substantive differences (see SI Section 6 and Fig. S7).
9	As an aside, I found the label "social norm followers" a bit awkward b/c the	According to Bicchieri's definition (see p. 11)², social norm followers are only those individuals who are conditionally responsive to both

	other strategies were also norm followers.	empirical and normative expectations. This is why we refer to types influenced by both empirical and normative expectations as social norm followers. We have also specified that these are types who place a similar weight on empirical and normative expectations in determining their contribution behaviour. We have clarified this aspect of Bicchieri's definition (p. 2, ll. 57, 60-62; and in sections 4 and 5 of the SI). and we refer back to these criteria in our classification of types (p. 8).
10	Page 12 of the supplemental. Do you mean accuracy rather than consistency in this discussion?	Thank you for pointing this out. We have corrected it.

REVIEWER 2

#	Question/Comment	Answer
11	To me, the most striking finding is one that actually seems to work against the authors' conclusions: their own figure 3b and associated text. Roughly 75% (!!) of the participants more or less ignore their own normative expectations when deciding how much to contribute. This seems bad for an account based on norms. The single largest subsample, 37% are impervious to both normative and empirical expectations. 38% of players based their behavior in some way on their empirical expectations – and only empirical expectations (some contributing more when others do, some trying to contribute just enough as a group to meet the threshold). Broadly, this latter set is consistent with cooperation in a coordination game (which the present game is), regardless of whether there are norms involved. Perhaps I am misunderstanding something, but the authors would need a convincing explanation for why these results don't actually work against their view of the importance of norms. There are many non-normative reasons empirical expectations would matter in a coordination game.	We used the k-means clustering algorithm with five clusters to allocate subjects to our different types. This algorithm finds clusters of subjects that are most similar to each other, which, in our context means that only subjects substantively influenced by normative expectations are included as either social norm followers or normative co-operators. We adopted this approach as it provides a stringent categorisation of types. In other words, we did not want to claim that all subjects who are influenced to some extent (i.e. >0) by normative expectations are normative or social norm types. Thus the 25% of subjects the reviewer refers to can be seen as a lower bound estimate. When we use a less stringent but simpler categorisation approach, and we partition the behaviour space into four quadrants (top left, top right, bottom left, and bottom right), we find that the majority of our subjects (77.2%; 213/276) are positively influenced to some extent (i.e. >1) by normative expectations. We have added the above points to p. 8 in the behavioural types classification section. Another way to see the strong role of normative expectations in shaping behaviour is from Figure 3 panel A. Average contribution under Low EE and Low NE is 33.9 while under Low EE and High NE it is substantively larger at 48.7. As a further robustness test, we have run another classification of typology by implementing an alternative way to measure empirical and normative expectations influence. The results of this analysis remain consistent with the initial one in terms of the proportion of behavioral types.

		The discussion and additional analysis are presented in the Section 6 of the SI.
12	Related, Figure 3b needs to be re-drawn. The authors visually privilege their hypothesis by stretching the normative axis relative to the empirical axis. (I.e., it takes twice as much change on the empirical axis to look like as big as an identical change as on the normative axis.)	Thank you for point this out. We have compressed the y-axis of Figure 3B to be equal to the x-axis. The elongation of the y-axis, relative to the x-axis, was unintentional and was due to the two-panel nature of the figure (coupled with the default figure combination settings in Stata).
13	Theoretically, why are groups reshuffled every round? Typical discussions of norms most centrally focus on norm developing within a group. But the current design is more like a norm developing within a meta-population. I'm not sure if this a problem but would like to know the authors' thoughts.	We agree that most discussions and experiments on social norms adopt a partner matching protocol instead of a quasi-stranger matching (random matching in larger groups with no identifying information) one as we have done in this work. The two protocols however fit two quite different scenarios. With a partner matching protocol, one is interested in the emergence of norms regulating behavior in long term relations developing in small-scale groups while, using the stranger matching protocol, one is interested in norms that influence recurrent but anonymous decision settings in larger social contexts. These are two very distinct domains. The latter has been explored mostly theoretically (see for instance^{8,9} and in contributions adopting evolutionary game theory like Sugden¹⁰ and Young¹¹) and less so experimentally (but see¹²). Since the tightness-looseness theory is mostly focused on norms that regulate interaction among “strangers”, we decided to use the stranger\random matching approach for this study. People typically face a series of cooperative decisions in multiple small groups throughout their day (e.g. working with colleagues, trading with strangers in a market) and based on these small-group interactions infer a more general population level norm that they apply to future small group interactions.

		We have now explicitly justified our design choice on p. 3, ll. 121-124.
14	I do not believe the authors do a direct test of one of their main hypotheses: that the rate at which norms change is faster in low-high than in high-low. They show that there is a change in norms when there is a change in risk in low-high and they show that there is no change in norms when there is a change in risk in high-low. But they do not do the direct “difference in difference” test.	We thank the reviewer for highlighting this. We have now tested the difference between the treatments directly in two ways. First, we compare whether the one round change in contributions (from rounds 14-15) is different across the two treatments relative to their “baseline” comparisons (round 14 from High Low is the baseline for round 15 Low High and round 14 from Low High is the baseline for round 15 High Low). This tests whether contributions in one treatment match closer to their comparison baseline than in the other treatment. Second, we compare whether the contributions change over the rounds (in rounds 15-28) following the initial response by comparing the regression slopes against each other. We find that contributions in Low High are closer to their target level than contributions in High Low by round 15, indicating that there is resilience in the latter. We also find that, contributions slowly decline over time (in rounds 15-28) in High Low to the baseline level while Low High remains stable at the level that is comparable to its baseline. We state this in the manuscript on p. 6, ll. 191-95 and provide details in the SI in Section 3 and tables S11 and S12.
15	Norm strength is an incredibly important variable in the paper, but readers aren’t given an intuitive sense of what a particular value means. Norm strength is .71 in the low risk condition. Is there anyway to interpret that absolute number as meaningful or is the metric purely arbitrary? This is relevant because the high risk condition had norm strength of .78. The difference was significant, but given how much power this study had (given a	We think that this is a very good point. To help interpret the scale of norm strength, we have calculated a random decision baseline value that reflects what would be observed if our subjects were making their decisions randomly for contribution, empirical expectations, normative expectations, and personal normative beliefs. We identify this value (0.424) using simulations with 1000 repetitions and we have added it to Figure

	reasonable sample size and many, many repeated observations), it might be more relevant to see the effect size of the difference, either in a standardized metric like Cohen’s d or some sort of interpretation that makes the raw difference of .07 substantively meaningful. The authors point to the differing success of the groups as relevant to showing the substantive importance of the difference in norm strength. But I think raw success is largely beside the point. If the risk level was 50% probability of disaster, risk-neutral players would be indifferent between contributing to meet the threshold and not bothering at all. The actual 60% risk in the low risk condition isn’t much bigger than 50%. So there may be many players who, through some combination of low risk aversion and low confidence others will contribute enough, don’t bother to contribute. It might have nothing to do with norm strength.	2A and to its legend, in the manuscript text (p. 4, ll. 156-160) and p. 17 of the SI. Additionally, we have provided the Cohen’s d value (0.753) for the difference in social norm strength between the treatments (p. 4, l. 156). Finally, as a way to clarify the dynamics of its separate components, we have added a figure to the SI that plots consensus, accuracy, and specificity separately (Fig. S1). We thank the reviewer for raising this important point concerning risk. We now realise that an explicit test of the association between norm strength and group success was missing. To test more thoroughly our claim—that independently of risk probability stronger social norms support the ability of groups to reach the threshold—we have conducted a logistic regression in which the outcome is the log odds of reaching the threshold and the predictors are social norm strength and disaster risk (see Table S9 and Fig. S3). If the reviewer’s suggestion is correct, we would expect no association between social norm strength and the success of a group. Instead, we find that social norm strength is positively associated with reaching the threshold (OR=4.800, s.e.=2.550, p=0.003). We have added this in the manuscript text (p. 4, l. 163-164).
16	I was confused by a lot of the specifics in involved the norm strength measure and the component parts that make it up. 1) Most generally, it seems to hide a lot of information to simply multiply together the components of consistency, accuracy, and specificity. Shouldn’t those be analyzed separately?	From a theoretical perspective it is important to analyse consistency, accuracy, and specificity together. The reason is that in our view, if any one of these factors is 0, then a social norm cannot be said to exist irrespective of the strength of the others. If expectations are not consistent or accurate, then, respectively, there is no coordination in expectations and they do not reflect reality. In both cases coordinated, expectation-driven behaviour is precluded.

		While if a norm is unrestricted (0 specificity) then there is no content to it in the sense that all behaviours are acceptable. In our view, when there is no consistency, or no accuracy, or no specificity, arguably, a norm ceases to exist. Conversely the strongest norms are highly consistent, accurate, and specific. On p. 4, ll. 138-139 of the manuscript we point interested readers to Section 5 of the Supplementary Information where we explain this reasoning. Additionally, we include a figure of the separate values for accuracy, consistency, and range in the Supplementary Information (Fig. S1). This shows that each component displays the same pattern that find for the overall measure of norm strength.
17	2) Related, multiplying measures together to arrive at a final DV to analyse is usually considered a poor approach because multiplying random variables radically increases the standard deviation of the resulting variable. On the other hand, the CIs for norm strength by condition show that these variables are surprisingly well estimated	We multiply the factors together for the above reason (response to comment #16). The variation of social norm strength does not substantially increase since each of the three constituent variables are already normalised. We now mention that consensus, accuracy, and range are all normalised (p. 4, ll. 148- 149).
18	3) Many issues were unclear in the computation of consistency, accuracy, and specificity. Each computation is supposed to compute for a specific player, p , their score. So why is there a summation for p in the formula? Either it's just cluttering the formula and represents a "summation" of 1 element or something else is wrong or confusing (to me, at least).	Thank you for the comment and we agree that there were some typos in parts of our explanation and formulas. In the paper the three components are computed at the group level not at individual level. We have now corrected the typos in the text and made the formula hopefully more understandable (Section 5 of SI)
19	4) Similarly, why distinguish between group g and time t ? Participants are in a new group every time period, so aren't time and group effectively synonymous?	We have removed group g from all formulas to avoid confusion and used time t instead of round r . Now groups are referred as G_p to indicate the group of participant p at a given time.
20	5) Is there any reason besides simplicity that EE and NE are weighted identically?	We do this for simplicity; we have no prior reason to weight them differently.
21	6) More generally, why are empirical and normative elements simply added together?	According to the definition of social norms that we adopt (from Bicchieri ²), both

	They are different concepts. If a researcher measured personality traits of agreeableness and extraversion, it does not seem to be coherent to add those (raw) scores together and say you've created a general 'personality' score. (Perhaps you could standardize each score first and then aggregate for a "positive valence" personality score.) Are empirical and normative expectations etc different?	empirical and normative expectations are necessary to claim the existence of a social norm. For this reason, we need to consider both in our metrics (consistency, accuracy, and specificity). It is also worth noting that empirical and normative expectations are normalised before being combined.
22	7) That R subscript symbolizes rank number for a particular (e.g.) expected contribution is not formally stated; readers are left to infer it.	We have made it clearer that contributions and expectations are ranked according to R in Section 5 of the SI (p. 15, l. 531).
23	8) It's never explained why the formulas include a normalizing max term in the denominator.	Group sizes can differ due to dropout or inactivity. Thus it is important to normalise the factors to account for these temporary differences in group size. We now explain this on p. 4, ll. 148-149.
24	9) Pairing things up for deviances by ranking two separate lists seems potentially arbitrary. Consider a person whose empirical estimations were (90, 80, 70, 60, 50) and the actual other group members contributed (100, 90, 80, 70, 60). Pairing up by ranks creates a uniform set of deviances of 10, yet just looking at the numbers suggest the person was quite accurate for 4 out of 5 group members, but they radically misestimated a single player, the player who actually contributed only 50. On the other hand, I played around with some sets like these and got the same set of absolute deviances regardless of whether I computed, taking this example, (90, 80, 70, 60, 50) - (100, 90, 80, 70, 60) or (90, 80, 70, 60, 50) - (90, 80, 70, 60, 100). In both cases the sum of the variances is 50. However, the latter version had greater variance in its deviances. So I'm not sure this is actually a problem but the authors should say more on why this is the appropriate strategy.	We elicited EE and NE using a ranked method. Since there were no identifiers among subjects and there was re-shuffling after each round, they could not associate estimates with specific individuals. Instead, we asked subjects to report their empirical expectations and normative expectations in a rank ordered way e.g. highest top and lowest at the bottom. We thus use the ordering that subjects provided us.
25	10) A similar question arises in the specificity measure. Why are participants' *individual* estimates for other players compared to the group average? Why not	We use specificity to indicate the amount of variation in acceptable behaviours such that the greater the variation in acceptable behaviours the lower the specificity.

the participants' average estimates to the group average? In part this measure, as computed, could just be measuring attentiveness or thoughtfulness. Consider two players. Both have identical, perfect estimates of the average mind of their other group members. But one is lazy and just puts this average estimate down for all 5 group members. The other one thinks hard and tries to come up with a reasonable distribution around the average for the other group members. The lazy person will have perfect specificity merely because they are lazy. Perhaps this is an extreme case, but nonetheless more needs to be said about the specific measure's computational form.	If we were to compare the average (as opposed to the distribution as we currently do) of each individual's expectations to their group's average, we would lose a lot of useful information. More concretely, the measure would mix situations in which people believe there are two (or more) norms within a group and those that believe there is a single norm. Consider the following as an illustration. Imagine that a group's average empirical expectation is 50. Individual i in that group reports her empirical expectations as (0, 0, 50, 100, 100) while individual j, also in that group, responds with (50, 50, 50, 50, 50). If we use the averaging approach, then both individuals receive a score of 0 and are thus interchangeable (since the group average is 50 and the average expectation for each is also 50). Conversely, with our approach i receives a score of 200 ($0-50 + 0-50 + 50-50 + 100-50 + 100-50$) while j receives a score of 0 ($50-50, 50-50, 50-50, 50-50, 50-50$) allowing us to separate between these dramatically different situations. We mention this point in the Supplementary Information (p. 17, fn. 1). Regarding the reviewer's second point about attentiveness. We don't believe that specificity simply measures attentiveness. The reason is that we elicited expectations in an ordered way and we incentivised them based on this ordering. That is, subjects knew that we would compare their ordered list of responses to the ordered list of true values for their group (empirical expectations to contributions and normative expectations to personal normative beliefs) and we would compare the two to calculate how much they earned. The closer they matched, the more they earned.
---	--

		We have added a section in the SI to explain our elicitation methods (Section 4).
--	--	---

REVIEWER 3

#	Question/Comment	Response
26	The motivation and literature discussion seem disconnected from what the experiment does: (1) formation of cultures is a long process, the dynamics of which cannot be captured in a 30-day time window.	We agree that the formation of cultures is a long-term process. Our paper however focuses only on one important component of cultures that is its social norms. Indeed, social norms can sometimes change quickly. Consider, for instance, how rapidly established social norms like shaking hands shifted, or new ones emerged, like wearing masks, since the start of the COVID-19 pandemic. Because social norms are a key part of cultures, we also believe that studying social norm can contribute to our understanding of broader cultural dynamics. By testing whether social norms change in response to simulated risk, we are testing a key prediction of the tightness looseness theory of culture change: that social norms become stronger in response to threat^{1,4}. If other social norms were to similarly change due to threats, this may have potential consequences for a more general cultural shift. We have clarified what we are doing in the text (p.1, ll. 43; p. 2, ll. 51-52).
27	(2) those changes have been the result of one-off exogenous shocks, whereas we are still in the midst of the pandemic. At the least, the study would need to examine post-pandemic behavior to make a strong claim about persistent changes in social norms.	The changes in our experiment do not come about from a one-off shock in risk. Once the risk probability changes at round 15 it becomes a stable feature of the new environment. Thus subjects are exposed to the shock for 14 rounds (corresponding to 14 days). We have clarified this in the text (p. 4, ll. 130-132). Although we agree with the reviewer that studying post-pandemic behaviour is an interesting direction of future research, it is important to clarify that our experiment was conducted before the current COVID-19 pandemic (data collection finished in September 2018). And, in the context of our experimentally increased risk, we do study behaviour both pre and post risk change.
28	Relatedly, the authors emphasize their ability to study behavior in the context of a disaster, but in reality, the authors implemented a repeated threshold	We implement a one-round version of the collective-risk social dilemma that was specifically designed to model a natural disaster (dangerous climate change originally) ¹³ . Since

	Public Goods Game with framing. To make broader, externally valid claims, you'd need to follow the existing research on natural disasters to capitalize on the impact of actual disasters with big implications for society.	then, many other studies, both empirical and theoretical, use variants of the collective-risk social dilemma to study natural disasters¹⁴⁻¹⁷. Regarding the external validity of our experiment, we aimed to increase this relative to standard laboratory studies by (i) using an “extra-laboratory” approach¹⁸ in which subjects made decisions in their normal environment and during their everyday lives (instead of in the focused and sterile laboratory), (ii) having the experiment last for 30 days instead of the usual one or two hours, and (iii) recruiting a more diverse range of subjects than is often used. This last point can be seen by the fact that 50% of our subjects are not students and the average age is 30 years. There is also now a growing literature testing the external validity of laboratory, or extra-laboratory experiments, and the results so far are encouraging suggesting that these types of studies can, at least in some cases, translate into “real world” behaviour¹⁹⁻²⁷. To mention one example, Rustagi and colleagues²⁸ show that groups in Ethiopia with more conditional cooperators, as measured using public goods game, are better able to manage their actual forest commons. Finally, concerning the statement “the need to follow the existing research on natural disasters to capitalize on the impact of actual disasters with big implications for society”. Although we are interested in providing causal evidence about the effect of risk on social norms and their strength, our research was motivated by a body of research using observational data on natural threats; for instance, Gelfand et al.’s original study on tightness looseness⁴ but also others^{1,29,30}. These studies provide the real-life context for our experiment, but, they do not show the causal role of disasters on social norm change. Our aim here is to pin down the causal relationship between the two.
29	In terms of design, there is too much going on (Fig. 1 is everything but intuitive) and the experiment	We have re-examined closely Figure 1 and agree that it was unclear. We have re-designed it with the aim of increasing clarity.

	description is rather confusing. For example, the authors state that participants participated in two experimental sessions in June and September 2018. It took me a while to realize that this is not an indicating of a within-design (participants doing one experiment in June and coming back in September), but rather an indication that the authors collected data in two waves with different participants.	We have clarified that this is a between subjects design in the manuscript and that subjects participated in one of the two sessions (p. 3, ll. 113, 114).
30	On this note, however, a lot of important information is missing: what about attrition over the course of 30 days and how was this accounted for in the analyses (e.g., one can imagine that certain types of people drop out along the way and thus introduce noise into the results)?	We thank the reviewer for pointing out this important concern. We deal with attrition in the following ways. First, we accounted for attrition in the design of our experiment in such a way as to minimize the effect of attrition on group dynamics. Re-shuffling groups each round meant that we could group together excluded participants (those who missed the first day or 4 or more decisions over the remaining 29 days) with each other in excluded groups. We also used a procedure for reducing the effect of these dropout on the active groups: an inactive subject contributed a randomly chosen amount from an active participant. So, these contributions are a reflection of the group's decisions (naturally, we excluded inactive decisions from the analyses). We mention this point on pp. 3, 4, ll. 120-125. Second, we can check whether the sample composition within each treatment differs from round 1 to round 28. If the sample composition remains constant throughout the experiment, then this suggests that subjects are randomly dropping out and thus attrition does not bias our estimates. We test this by running 26 unpaired t-tests (13 for each treatment) comparing sample composition on round 1 to round 28 on 13 variables. We deliberately do not correct for multiple comparisons (e.g. using Bonferroni correction) in order to provide an upper bound of differences. We find no significant nor substantive differences

		in sample composition between round 1 and round 28 within each treatment. The p-values for the 13 comparisons for High Low have mean=0.924, s.d.=0.049, min.=0.837, and max.=0.985 while the 13 comparisons for Low High have mean=0.713, s.d.=0.112, min.=0.480, and max.=0.932. We mention this in the Supplementary Information (p. 21). We also mention both points in the manuscript on p. 6, ll. 225-230. It is also worth keeping in mind, however, that attrition over the course of the experiment is low at 8% (23/286). Moreover, the number of missing observations (due to either dropout or inactivity in some rounds without dropout) was similarly low. For instance, 7% (567/8008) contribution decisions are missing.
31	Since data was collected with several months apart but still pooled in the analyses, it would be important to show that these participants do not differ along relevant dimensions.	We agree with the reviewer about the need for this additional analysis. We follow the same procedure as in the above response (response to comment #30). To check whether there are differences between our samples in the High Low and the Low High treatments, we conducted a series of 26 unpaired t-tests comparing the samples at round 1 in the two treatments (13 for each). We deliberately do not correct for multiple comparisons (e.g. using Bonferroni correction) in order to provide an upper bound of differences. Despite this, none of the comparisons are significant at the 5% level (distribution of p-values: mean=0.351, s.d.=0.214, min.=0.060, and max.=0.691) although one, for experience with experiments, is close at $p=0.060$. Yet even if this is treated as significant, it is most likely to be noise. This is because if there were systematic differences, they should be reflected in multiple variables and not only in this arbitrary one. Another reassuring observation— suggesting that participants did not differ in important dimensions—is that social norm strength is very similar across the two groups in rounds 1-3. And, it takes multiple days before the treatment effects

		on social norm strength emerge. If participants were different in important ways this should not be case and social norm strength could differ already from round 1. We briefly mention the former point in the manuscript on p. 6, ll. 230-232. Additionally, we elaborate on this point in the Supplementary Information (p. 20).
32	More generally, the extent to which the experimental design (reshuffling groups after each period) is consistent with the theoretical framework that the authors put forward (social norms are properties of reference networks) is unclear. If you want to study how norms change and develop, why would you want to shuffle the groups every time? Asking for beliefs at the ‘global’ rather than specific group level does not resolve this tension.	Please see our response to comment #13 of Reviewer 2. We have now explicitly justified our design choice on p.3, ll. 121-124.
33	I had a hard time following the authors’ arguments about the chain of causality in their repeated game: both norm-related beliefs and actual behavior (with feedback!) are essentially elicited simultaneously. After day 1, the results are largely affected by endogeneity. Because feedback is given, participants are updating their beliefs about the environment, even if they are randomly matched with other participants next time (although no perfect stranger matching is attainable given the group sizes and number of periods). The authors need to be more explicit about the causal inferences that they believe can be drawn from this design.	We thank the reviewer for this important comment. Concerning endogeneity, we agree that in terms of subjects’ beliefs and contributions after day 1 there is a feedback loop. Indeed, the co-dynamics of contribution and expectations are one of the main results that we wanted to capture in the paper. Although we find that social expectations explain substantial variation in contributions (the two variables alone account for 22% of the variance in contributions), the reviewer is also correct that, as a consequence of feedback, we cannot draw clean causal inferences about the role of expectations in shaping behaviour during the standard game play (as in Table 1). We have made this clear on p. 6, ll. 232-233. Yet there are two kinds of causal inferences that we can make with this experiment. First is the difference in social norm strength, expectations, and contribution behaviour between the risk probabilities (high or low) and between the different ordering of the risk probabilities (i.e. the treatments). Since subjects are assigned

		exogenously to these conditions, there should be no plausible confounds. This allows us to test the relevant research questions cleanly. The second kind of causal inference concerns the “conditional contributions” of subjects in response the empirical and normative expectations that we changed in an additional stage on specific rounds. To clarify, on rounds 1, 5, 10, 14, 15, 19, 24, and 28 of the collective-risk social dilemma we asked subjects to report their contribution in each of four cases:  1. the majority of your group members put in at least 50 points and believe that you all should spend at least 50 points. 2. the majority of your group members put in less than 50 points and believe that you all should spend at least 50 points. 3. the majority of your group members put in at least 50 points and believe that you all should spend less than 50 points. 4. the majority of your group members put in less than 50 points and believe that you all should spend less than 50 points. They also knew that one of their responses would be implemented based on the actual beliefs and contributions of the other members of their group in that round and thus the elicitation was incentive compatible. (We elicited this before they received feedback from that round). Put differently, we changed their expectations and measured subjects’ responses. We find that subjects contributions are strongly affected by their empirical and normative beliefs (see Figure 3A and p. 8) and that based on their responses to their categories we can identify different behavioural types (Figure 3B and p. 8).
34	The analysis was also rather confusing at times: for example, the pre-registration is powered for an alpha of 0.05, so it seems rather odd to refer to p-values above that as significant (e.g., line 140ff).	We believe that the reviewer is referring to the p-value of 0.071 (now on p. 6, ll. 201). We do not refer to this p-value as significant. Having said that, we do interpret it as being consistent with stronger social norms for two reasons.

		First, the difference in punishment strength is consistent with expectations of differences in punishment strength (analysis now added to p. 6, ll. 202-204). Second, our general approach is to interpret p-values as a continuous and not dichotomous construct. By dichotomizing our results—and simply claiming significance or not below or above a threshold—we lose information and present an overly-deterministic picture. Our approach is consistent with the American Statistical Association’s recent statement on p-values³¹ in which they recommend against dichotomizing as a general principle (see Principle 3): “Scientific conclusions and business or policy decisions should not be based only on whether a p-value passes a specific threshold. Practices that reduce data analysis or scientific inference to mechanical ‘bright-line’ rules (such as ‘$p < 0.05$’) for justifying scientific claims or conclusions can lead to erroneous beliefs and poor decision making.”
35	On the other hand, Figure 3 illustrates a breakdown by low/high EE/NE, but I was unable to find a precise definition of what constitutes low and high. Without it, this classification appears rather arbitrary.	We thank the reviewer for raising this point and we now realise that the breakdown and elicitation were unclear. We have clarified the elicitation method in the Supplementary Information (sections 4 and 6) and direct readers to it on p. 8, l. 268 of the manuscript. Based on the elicitation method, this is the only reasonable classification for displaying conditional contributions in our setting. In Figure 3A, we plot the average of subjects’ responses to each of the four combinations of expectations.
36	Lastly, in contrast to the existing social preference literature that also includes classifications of free-riders, unconditional cooperators etc., the authors find ‘new’ types. It remains unclear what one can learn from this new classification as now distinct insights or policy implications are provided at any point.	Our contribution concerning the identification of new types is two-fold. Partly it is fundamental. Much work posits that conditional cooperators are driven by social norm considerations⁵⁻⁷, yet, no study that we know of has directly tested this assumption by manipulating the two kinds of expectations (empirical and normative) that have been argued to matter for social norm compliance. We do so here and show that only a subset of conditional cooperators are truly motivated by social norms. We explain our contribution more clearly in the manuscript (p. 8, ll. 269-272).

		Our other contribution on this topic is that it hints at policy approaches. We mention this conjecture in the final paragraph of our manuscript where we argue that policies aimed to shift collective action should focus more on shifting normative messaging than empirical messaging since empirical messages can increase free-riding (as shown by the threshold type) (see p. 9, ll. 309-317 of the manuscript).
--	--	---

REFERENCES

1. Roos, P., Gelfand, M., Nau, D. & Lun, J. Societal threat and cultural variation in the strength of social norms: An evolutionary basis. *Organizational Behavior and Human Decision Processes* **129**, 14–23 (2015).
2. Bicchieri, C. *The grammar of society: The nature and dynamics of social norms*. (Cambridge University Press, 2006).
3. Price, R. H. & Bouffard, D. L. Behavioral appropriateness and situational constraint as dimensions of social behavior. *Journal of Personality and Social Psychology* **30**, 579–586 (1974).
4. Gelfand, M. J. *et al.* Differences Between Tight and Loose Cultures: A 33-Nation Study. *Science* **332**, 1100–1104 (2011).
5. Fischbacher, U., Gächter, S. & Fehr, E. Are people conditionally cooperative? Evidence from a public goods experiment. *Economics Letters* **71**, 397–404 (2001).
6. Fehr, E. & Gintis, H. Human motivation and social cooperation: experimental and analytical foundations. *Annual Review of Sociology* **33**, 43–64 (2007).
7. Fehr, E. & Fischbacher, U. Social norms and human cooperation. *Trends in Cognitive Sciences* **8**, 185–190 (2004).
8. Kandori, M. Social Norms and Community Enforcement. *The Review of Economic Studies* **59**, 63–80 (1992).
9. Okuno-Fujiwara, M. & Postlewaite, A. Social Norms and Random Matching Games. *Games and Economic Behavior* **9**, 79–109 (1995).
10. Sugden, R. *The Economics of Rights, Co-operation and Welfare*. (Basil Blackwell, 1986). doi:10.1057/9780230536791.
11. Young, H. P. *Individual strategy and social structure: An evolutionary theory of institutions*. (Princeton University Press, 1998).
12. Camera, G. & Casari, M. Cooperation among Strangers under the Shadow of the Future. *American Economic Review* **99**, 979–1005 (2009).
13. Milinski, M., Sommerfeld, R. D., Krambeck, H.-J., Reed, F. A. & Marotzke, J. The collective-risk social dilemma and the prevention of simulated dangerous climate change. *PNAS* **105**, 2291–2294 (2008).
14. Chakra, M. A., Bumann, S., Schenk, H., Oschlies, A. & Traulsen, A. Immediate action is the best strategy when facing uncertain climate change. *Nat Commun* **9**, 1–9 (2018).
15. Milinski, M., Hilbe, C., Semmann, D., Sommerfeld, R. & Marotzke, J. Humans choose representatives who enforce cooperation in social dilemmas through extortion. *Nat Commun* **7**, (2016).

16. Vicens, J. *et al.* Resource heterogeneity leads to unjust effort distribution in climate change mitigation. *PLOS ONE* **13**, e0204369 (2018).
17. Tavoni, A., Dannenberg, A., Kallis, G. & Lösschel, A. Inequality, communication, and the avoidance of disastrous climate change in a public goods game. *PNAS* **108**, 11825–11829 (2011).
18. Charness, G., Gneezy, U. & Kuhn, M. A. Experimental methods: Extra-laboratory experiments-extending the reach of experimental economics. *Journal of Economic Behavior & Organization* **91**, 93–100 (2013).
19. Reindl, I., Hoffmann, R. & Kittel, B. Let the others do the job: Comparing public good contribution behavior in the lab and in the field. *Journal of Behavioral and Experimental Economics* **81**, 73–83 (2019).
20. Cohn, A. & Maréchal, M. A. Laboratory Measure of Cheating Predicts School Misconduct. *The Economic Journal* **128**, 2743–2754 (2018).
21. Cohn, A., Maréchal, M. A. & Noll, T. Bad Boys: How Criminal Identity Salience Affects Rule Violation. *Rev Econ Stud* **82**, 1289–1308 (2015).
22. Gächter, S. & Schulz, J. F. Intrinsic honesty and the prevalence of rule violations across societies. *Nature* **531**, 496–499 (2016).
23. Dai, Z., Galeotti, F. & Villeval, M. C. Cheating in the Lab Predicts Fraud in the Field: An Experiment in Public Transportation. *Management Science* **64**, 1081–1100 (2017).
24. Armantier, O. & Boly, A. Comparing Corruption in the Laboratory and in the Field in Burkina Faso and in Canada. *The Economic Journal* **123**, 1168–1187 (2013).
25. Potters, J. & Stoop, J. Do cheaters in the lab also cheat in the field? *European Economic Review* **87**, 26–33 (2016).
26. Hanna, R. & Wang, S.-Y. Dishonesty and Selection into Public Service: Evidence from India. *American Economic Journal: Economic Policy* **9**, 262–290 (2017).
27. Fehr, E. & Leibbrandt, A. A field study on cooperativeness and impatience in the Tragedy of the Commons. *Journal of Public Economics* **95**, 1144–1155 (2011).
28. Rustagi, D., Engel, S. & Kosfeld, M. Conditional cooperation and costly monitoring explain success in forest commons management. *Science* **330**, 961–965 (2010).
29. Harrington, J. R. & Gelfand, M. J. Tightness–looseness across the 50 united states. *PNAS* **111**, 7990–7995 (2014).
30. De, S., Nau, D. S., Pan, X. & Gelfand, M. J. Tipping Points for Norm Change in Human Cultures. in *Social, Cultural, and Behavioral Modeling* (eds. Thomson, R., Dancy, C., Hyder, A. & Bisgin, H.) 61–69 (Springer International Publishing, 2018). doi:10.1007/978-3-319-93372-6_7.
31. Wasserstein, R. L. & Lazar, N. A. The ASA Statement on p-Values: Context, Process, and Purpose. *The American Statistician* **70**, 129–133 (2016).

REVIEWER COMMENTS

Reviewer #1 (Remarks to the Author):

I appreciate the authors' responsiveness to the queries from the first round of reviews and the manuscript is much improved. I believe the work provides an important contribution in illustrating the causal impact of collective risk on cooperation over time. There are a number of issues that still warrant attention:

1. The summary should be more informative of the main findings. For example, "considering different orders or high or low risk" is quite vague. I would provide the key insights that the study provides in this paragraph.

2. Page 1. What do you mean by "Inefficient social norms"

3. The order of the hypotheses is disconnected from how the results are discussed which makes the manuscript hard to follow. In the hypothesis section, I would suggest starting with the main focus of the experiment which is on collective risk and tightening (there is no direct hypothesis of this so this should be added). Then the rate of change (Hypothesis 4) would come next, as it does in the results section, and the hypotheses on social norms and contribution would come after these. I would note that the whole idea of whether groups can reach the threshold is not discussed in the introduction. As this is a key finding, I would suggest motivating this question much earlier in the manuscript.

4. It would be useful to connect the results on high/low and low/high and the results on social norms and contribution. In some ways, the paper seems like "two mini-papers" otherwise. For example, does condition (High-Low, Low-High) affect the predictive power of empirical and normative expectations as well as personal beliefs for contributions? One possibility is that personal beliefs predict less under high threat, for example. This would be consistent with research has found that values are less predictive of behavior in contexts of strong norms (See paper, Elster, A., & Gelfand, M. J. (2020). When guiding principles do not guide: The moderating effects of cultural tightness on value-behavior links. *Journal of Personality*).

5. The punishment data are still the weakest part of the paper given that they were only measured at the end of the study. I would suggest putting this in the supplemental.

6. The policy implications (page 9) are unclear. If I understand correctly, you're arguing that policy interventions should be developed to keep norm strength high even when there is low risk? Why would that be needed from an evolutionary perspective?

Reviewer #2 (Remarks to the Author):

I was a reviewer on the previous version of this article. The authors have satisfactorily answered my concerns and I would support publication of this article.

REVIEWER 1

#	Question/Comment	Response
1	The summary should be more informative of the main findings. For example, “considering different orders or high or low risk” is quite vague. I would provide the key insights that the study provides in this paragraph.	We agree and have re-written the summary.
2	What do you mean by “Inefficient social norms”	There is no generally agreed upon standard for inefficient social norms. To avoid confusion, we have removed the term “inefficient”; it was not essential to the point we make there (l. 43).
3	The order of the hypotheses is disconnected from how the results are discussed which makes the manuscript hard to follow. In the hypothesis section, I would suggest starting with the main focus of the experiment which is on collective risk and tightening (there is no direct hypothesis of this so this should be added). Then the rate of change (Hypothesis 4) would come next, as it does in the results section, and the hypotheses on social norms and contribution would come after these. I would note that the whole idea of whether groups can reach the threshold is not discussed in the introduction. As this is a key finding, I would suggest motivating this question much earlier in the manuscript.	Upon re-reading our manuscript with fresh eyes, we agree with the issue identified: the mismatch between the hypotheses and the results is confusing. We thank the reviewer for highlighting this issue. To solve it, we have re-ordered the results to match the hypotheses, in our view the more logically coherent sequence. We now begin by (i) establishing whether cooperative social norms exist in our setting and causally motivate behaviour, (ii) study heterogeneity in norm following by identifying multiple types, and (iii) we end by testing the between treatment effects of risk on the strength of social norms. We think that this re-ordering also solves comment 4 and partly solves comment 5. To avoid obscuring other changes, we do not highlight this re-ordering in the manuscript. We respectfully disagree with the reviewer about adding a hypothesis. The hypotheses we present in the manuscript are our pre-registered ones and we wish to remain with these. We have highlighted further in the introduction the point that groups with stronger social norms should be (and are) better able to solve the risks they face (ll. 47-50, 59).
4	It would be useful to connect the results on high/low and low/high and the results on social norms and	We agree with the reviewer’s concern. Our re-ordering of the results (see response to comment

	contribution. In some ways, the paper seems like "two mini-papers" otherwise. For example, does condition (High-Low, Low-High) affect the predictive power of empirical and normative expectations as well as personal beliefs for contributions? One possibility is that personal beliefs predict less under high threat, for example. This would be consistent with research has found that values are less predictive of behavior in contexts of strong norms (See paper, Elster, A., & Gelfand, M. J. (2020). When guiding principles do not guide: The moderating effects of cultural tightness on value-behavior links. Journal of Personality).	3) solves this issue as it creates a more logically coherent flow.
5	The punishment data are still the weakest part of the paper given that they were only measured at the end of the study. I would suggest putting this in the supplemental.	We respectfully disagree with the reviewer. Punishment is an additional indicator that social norms of contribution exist. Given this aim, it is a strength of our design that we only elicit punishment at the end of the experiment. This is because at this point subjects cannot influence others' behaviour in any way by using punishment (there are no further rounds of the collective-risk social dilemma). Consequently, instrumental motivations for punishing our ruled out. In contrast, normative considerations can motivate costly punishing behaviour. We now mention this on ll. 80-81, 85-86. Nevertheless, we do wish to highlight that our re-ordered results reduce the prominence of punishing since it is now integrated in other sub-sections.
6	The policy implications (page 9) are unclear. If I understand correctly, you're arguing that policy interventions should be developed to keep norm strength high even when there is low risk? Why would that be needed from an evolutionary perspective?	We have clarified our argument of the policy implications (ll. 310-314).

REVIEWER COMMENTS

Reviewer #1 (Remarks to the Author):

I applaud the authors on this great revision. All of my comments have been addressed and I think the flow of the paper is much better. Congratulations on a great paper.